# A directly comparative two-gate case–control diagnostic accuracy study of the pure tone screen and HearCheck screener tests for identifying hearing impairment in school children

Obioha C Ukoumunne,[1] Chris Hyde,[2] Mara Ozolins,[3] Zhivko Zhelev,[1]
Sam Errington,[3] Rod S Taylor,[2] Claire Benton,[4] Joanne Moody,[5] Laura Cocking,[6]
Julian Watson,[7] Heather Fortnum[3]

► Prepublication history and additional material is available. To view please visit the journal (http://dx.doi.org/ 10.1136/ bmjopen-2017-017258).

For numbered affiliations see end of article.

**Correspondence to**
Dr Obioha C Ukoumunne; O.C. Ukoumunne@exeter.ac.uk

## ABSTRACT

**Objectives** This study directly compared the accuracy of two audiometry-based tests for screening school children for hearing impairment: the currently used test, pure tone screen and a device newly applied to children, HearCheck Screener.

**Design** Two-gate case–control diagnostic test accuracy study.

**Setting and participants** Hearing impaired children ('intended cases') aged 4–6 years were recruited between February 2013 and August 2014 from collaborating audiology services. Children with no previously identified impairment ('intended controls') were recruited from Foundation and Year 1 of schools between February 2013 and June 2014 in central England. The reference standard was pure tone audiometry. Tests were administered at Nottingham Hearing Biomedical Research Unit or, for some intended cases only, in the participant's home.

**Main outcome measures** Sensitivity and specificity of the pure tone screen and HearCheck tests based on pure tone audiometry result as reference standard.

**Results** 315 children (630 ears) were recruited; 75 from audiology services and 240 from schools. Full test and reference standard data were obtained for 600 ears; 155 ears were classified as truly impaired and 445 as truly hearing based on the pure tone audiometry assessment. Sensitivity was estimated to be 94.2% (95% CI 89.0% to 97.0%) for pure tone screen and 89.0% (95% CI 82.9% to 93.1%) for HearCheck (difference=5.2% favouring pure tone screen; 95% CI 0.2% to 10.1%; p=0.02). Estimates for specificity were 82.2% (95% CI 77.7% to 86.0%) for pure tone screen and 86.5% (95% CI 82.5% to 89.8%) for HearCheck (difference=4.3% favouring HearCheck; 95% CI 0.4% to 8.2%; p=0.02).

**Conclusion** Pure tone screen was better than HearCheck with respect to sensitivity but inferior with respect to specificity. As avoiding missed cases is arguably of greater importance for school entry screening, pure tone screen is probably preferable in this context.

**Study registration number** Current controlled trials: ISRCTN61668996.

### Strengths and limitations of this study

► A public involvement representative was a full member of the study team and contributed to the development, conduct and interpretation of the study.
► The audiometry-based screening tests, pure tone screen and HearCheck Screener were directly compared in the same sample of children.
► The two-gate case–control study design used to identify cases and controls is known to be susceptible to bias.
► The estimates of accuracy of each test might be biased but estimates of comparative accuracy are unlikely to be affected by the design.

## INTRODUCTION

Identification of permanent hearing impairment at the earliest possible age is crucial to the development of speech and language and for ensuring the best opportunities for educational achievement and quality of life.[1] The highly sensitive and specific universal newborn hearing screen identifies the vast majority of children born with a hearing impairment.[2] Due to acquisition, progression or late onset of hearing impairment and geographical movement of families, however, a significant number of children remain to be identified with a permanent hearing impairment after the newborn period. The school entry screen (SES), a universal hearing screen when children start school, was established in 1955 and remains in place in many parts of the UK. It is considered a safeguard screen to identify hearing impairment.

The 2007 National Institute for Health Research (NIHR) Health Technology Assessment (HTA)-funded evaluation of the

cost-effectiveness of the school entry hearing screen in the UK[3] included a survey of practice which found that the audiometry-based pure tone screen (PTS) test[4] was used in all cases. The diagnostic accuracy studies identified by a review that was part of this evaluation found PTS to generally have higher sensitivity for minimal, mild and greater hearing impairments than alternative tests (tympanometry, otoscopy, transient-evoked otoacoustic emission tests, parent questionnaires, spoken word tests) for which evidence was identified.[3] These comparisons were, however, indirect and highly susceptible to confounding. Furthermore, most of these accuracy studies were undertaken in populations where the prevalence of undetected hearing impairment was considerably greater than that likely to be encountered in a system where a universal newborn hearing screening programme is in place. The estimates of accuracy were also based on small sample sizes. A relatively new device, HearCheck Screener (HC),[5] also audiometry based, came into the market in 2005 as a tool for screening for hearing impairment in adults in a general practice setting. It is less comprehensive and flexible than PTS but has the potential to be a quicker test in the school setting. It has not previously been assessed as a tool for screening children in the UK. For further details on how HearCheck Screener is used, refer to: http://www.connevans.co.uk/product/2831233/38SHEARCHECK/Siemens-HearCheck-Screener (accessed 18 May 2017).

The objective of this study was to compare the diagnostic accuracy of PTS and HC tests for hearing impairment of any type at or around school entry using full pure tone audiometry (PTA) as the reference standard.[6] The study was part of an HTA-funded programme of work with the wider aim of assessing the effectiveness and cost-effectiveness of the school entry hearing screen.[7] The full study protocol is available from the authors on request.

## METHODS
### Participants
This diagnostic test accuracy study used a directly comparative two-gate case–control design.[8]

### Intended cases
Hearing impaired children aged 4–6 years between February 2013 and August 2014 were identified by collaborating audiology services (centres) in central England. They had permanent sensorineural or conductive hearing impairment averaged across the four frequencies 0.5, 1, 2 and 4 kHz, either bilaterally (average of 20–60 dB HL) or unilaterally (any level ≥20 dB HL). Children were identified by the paediatric audiologist in each centre. The reference standard was PTA, and potential recruits were excluded if there was no record of a PTA in the previous 12 months or planned for the following 3 months, and the family was unwilling to travel to their local service or to Nottingham to undergo the assessment. Eligible children for whom parents provided agreement to take part were invited to undergo the two screening tests (PTS and HC),

either in their own homes or at Nottingham Hearing Biomedical Research Unit (NHBRU), depending on their preference.

### Intended controls
Children with no previously identified hearing impairment were recruited from the Foundation Year and Year 1 of schools in the Nottingham area (central England), between February 2013 and June 2014. The study researchers provided an agreed letter of invitation and information packs for the school. Children for whom agreement to take part was provided were invited to undergo the two screening tests and the PTA reference standard assessment at NHBRU.

### Procedures
Written informed consent was obtained from the parent or legal guardian before the child entered the study. Test data for all children and reference standard data for all intended controls were collected specifically for this study. For most of the intended cases the reference standard data were based on previous assessments otherwise unconnected to the study.

### Pure tone screen test
Headphones were placed over the child's ears and then pure tones presented across the key frequencies for speech understanding in the order 1, 2, 4 and 0.5 kHz. Each tone was held for 2 to 3 s, with staggered pauses. All four frequencies were tested in one ear before being tested in the other. To pass the screen in a given ear, the child needed to respond to two out of three presentations of each frequency at 20 dB HL to pass. The researcher was positioned to ensure they had a clear view of the child without giving any visual cues throughout the test. The child was instructed to place a ball onto a frame every time they heard a sound, however quiet. Hearing aids, glasses, hairbands and earrings were removed where relevant. A familiarisation tone (1 kHz at 60 dB HL) was presented to ensure the child had understood the instructions.

### HearCheck Screener test
The HC screener was placed over the child's ear and an automatic sequence of pure tones played at three levels at each of the frequencies 1 kHz (55, 35 and 20 dB) and 3 kHz (75, 55 and 35 dB). To pass the screen in a given ear, the child needed to respond to all six tones. The child indicated, usually by raising their hand, that they had heard each tone. The child was asked to remove hearing aids and also glasses and earrings if necessary for a good fit. A disposable cardboard ear cover was put into the HC for each ear. The HC was held against the first ear to be tested, often holding the child's head still with the free hand. The button was pressed and the first three tones were allowed to play for the first frequency. The button was then pressed again for the remaining three tones for the second frequency. The procedure was repeated on the other ear.

### Reference standard

After intended cases had their appointment, the researcher phoned the audiologist, asking them to post their most recent PTA results to NHBRU. For intended controls, PTA was carried out in NHBRU at the same session as the screening tests, using the audiometer in a sound-proofed booth with the child sat facing away from the equipment. PTA testing followed standard British Society of Audiology recommended procedure[6] without otoscopic examination or masking for air conduction only. Hearing impairment was considered present when the PTA reference standard threshold was ≥30 dB on at least one of the four frequencies (0.5, 1, 2 and 4 kHz) and considered absent when the reference threshold was <30 dB on all four frequencies.

### Other procedural details

Equipment was calibrated as per manufacturer's instructions. There was generally less background noise than would be expected in schools. The researchers were trained in administering the PTA and PTS by the audiologists in the Children's Hearing Assessment Centre in Nottingham, using a mixture of observation, practice on children and feedback.

The order of administering the two screening tests and which researcher undertook them was determined randomly. For intended cases, one researcher performed the PTS and another researcher performed the HC. For intended controls, one researcher carried out both the screening tests and then another researcher performed the PTA measurement. We sought to blind the second researcher to the results of the first test(s) by asking them to leave the room. The PTA result obtained from the audiologist for intended cases was examined only after the results of the screening tests were known.

Ethical approval was granted by the West Midlands, Staffordshire Research Ethics Committee (Ref: 106333).

### Statistical analysis

The target sample size was 80 hearing impaired children and 160 children with no hearing impairment. Eighty impaired children is large enough to estimate a sensitivity of 80% with a margin of error of 10.4% based on the lower bound of the 95% CI and 160 children without impairment is large enough to estimate a specificity of 80% with a margin of error of 7.0%. Accuracy was evaluated using the ear as the unit of analysis. In the main analysis, irrespective of intended case or control status, ears were defined as truly hearing impaired or not based on actual PTA reference standard results. Analyses were carried out using Stata statistical software V.13. We reported the absolute difference in percentages between the PTS and HC for each of sensitivity and specificity with 95% CIs and McNemar's test p value (using the Stata command mcc). We used analytical methods that recognise the correlation between results of ears belonging to the same child. Details of further exploratory analyses are provided in the see online supplementary appendix 1.

### Public involvement

The research question originates from a call from the NIHR HTA funding stream to evaluate the diagnostic accuracy of hearing tests and cost-effectiveness of school entry hearing screening programmes. We recruited JW, a parent of a child who has experienced conductive hearing impairment, to be a full member of the study team and an author on this paper. His input included comments on information literature for participating parents, development of methodology and the conduct of the study (eg, addressing recruitment challenges), attending study meetings and critical comments and suggestions on the final study report and this paper. We also included on our study steering committee a representative from the National Deaf Children's Society. Parents of participating children were offered the opportunity to receive a lay summary of the findings at the end of the study. Almost all parents took up the offer and it was sent to them.

## RESULTS

### Participants

Intended cases were recruited from 14 audiology services. We received 86 replies from 379 invitations sent by the audiologists. Eight children were ineligible, being outside the required age range. We were unable to contact one of the initial respondents, and we were unable to see a further two children due to researcher illness just before the close of recruitment. We recruited and tested the remaining 75 children (19.8% of those invited) (figure 1).

Intended controls were recruited from 51 of the 164 schools in the Nottingham area that were invited by post to take part. The 51 schools between them gave information packs to the parents of 2787 children, of whom 291 (10.4%) replied, confirming they would like to participate. Eight of the 291 children were subsequently found to be ineligible for the study (one was too old, six already

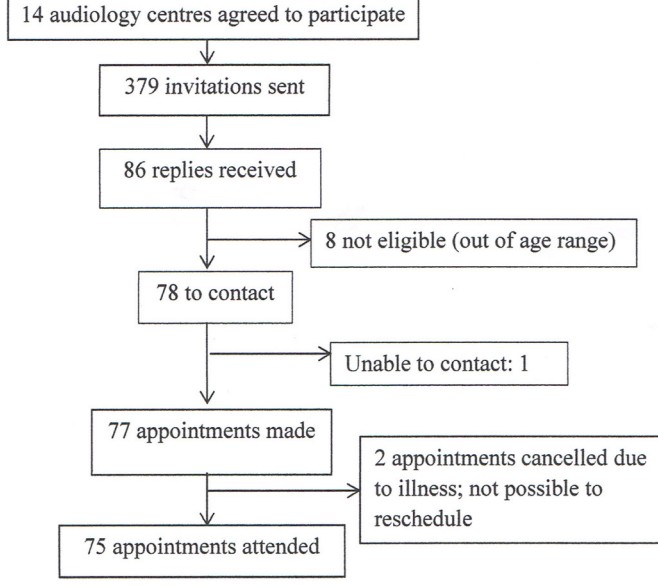

**Figure 1** Recruitment of intended case children.

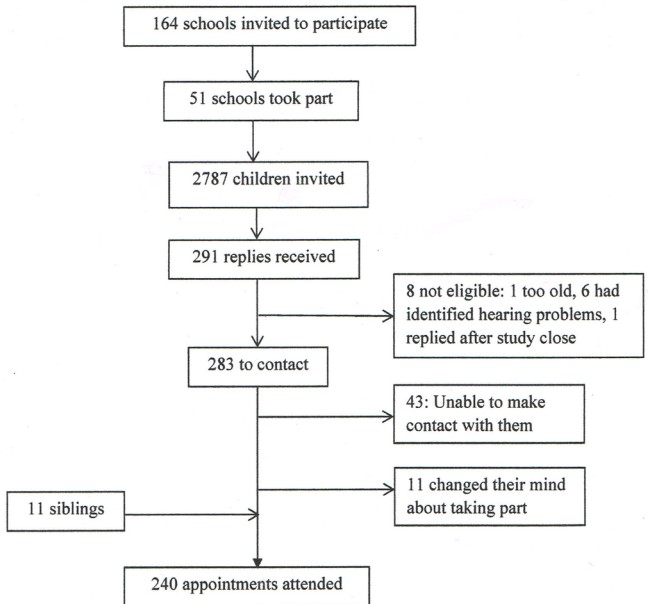

**Figure 2** Recruitment of intended control children.

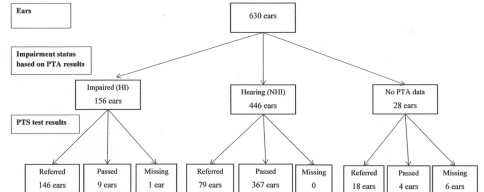

**Figure 3** PTS test results at ear level by hearing impairment status (PTA). PTA, pure tone audiometry; PTS, pure tone screen.

had hearing problems identified, one replied after recruitment closed), 11 changed their minds about taking part, and we were unable to see 43 either because we could not make an appointment (mostly not contactable) or they did not attend the arranged appointment. An additional 11 siblings of children who attended the appointment but who did not receive the invitation were in the correct age range and parents agreed for them to take part. The remaining 240 children were recruited as intended controls and seen for study appointments (figure 2).

Table 1 summarises the demographic characteristics of participating children by whether they were recruited via audiology services (intended cases) or via schools (intended controls). The groups were similar with respect to gender and age.

Intended controls completed the tests and reference standard on the same day. For intended cases, reference standard data were already available prior to the tests being administered for 65 children, and for the remaining 10 children, a reference standard assessment took place after administering the PTS and HC. The median time interval between reference standard and test assessment was 16 weeks. There were no adverse events from performing the tests and the reference standard.

### Number of ears with impaired or non-impaired hearing

Of the 630 recruited ears, 600 (95.2%) provided full data on the PTS and HC tests and scores for all four frequencies of the PTA reference standard and were included in the main analyses. Two hundred and ninety-five children provided full data on both ears and another 10 provided full data on just one ear. There were no indeterminate screening test or PTA results. The PTA reference standard categorised 155 ears as impaired and 445 as not impaired. The mean (SD) hearing level in dB at frequencies 0.5, 1, 2 and 4 kHz was 43.1 (21.0), 45.0 (22.5), 46.2 (25.0) and 49.0 (24.2), respectively, for impaired ears and 9.4 (7.4), 4.7 (7.5), 3.7 (6.6) and 4.9 (8.1), respectively, for hearing ears. One hundred and seven of the impaired ears belonged to children recruited from audiology services, and the remaining 48 ears belonged to children with no previously identified hearing loss.

Figures 3 and 4 present flow charts that describe the number of impaired ears (based on PTA ≥30 dB on at least one of the four frequencies) and hearing ears that passed and referred on the PTS and HC tests, respectively.

### Sensitivity and specificity

Table 2 summarises the relationship between the PTS and HC test results separately for impaired ears (first panel), hearing ears (second panel) and ears for which information on the reference standard was missing (third panel). The figures highlighted in bold in the first panel indicate the 155 impaired ears that were used

| Table 1 Demographic characteristics of children by recruitment source | | |
| --- | --- | --- |
| **Characteristic** | **Recruited via audiology services (intended cases) (N=75)** | **Recruited via schools (intended controls) (N=240)** |
| Male, n (%) | 38 (51) | 117 (49) |
| Age, mean (SD; range) | 5.4 (0.9; 3.9 to 7.0) | 5.4 (0.6; 4.0 to 6.9) |
| Ethnicity | | |
| White, n (%) | 61 (81) | 189 (79) |
| Black, n (%) | 2 (3) | 14 (6) |
| Asian, n (%) | 11 (15) | 10 (4) |
| Mixed, n (%) | 1 (1) | 22 (9) |
| Other, n (%) | 0 (0) | 5 (2) |

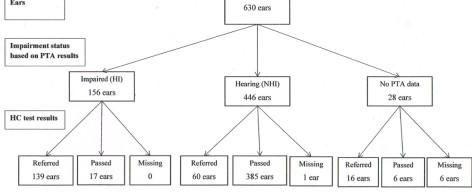

**Figure 4** HC test results at ear level by hearing impairment status (PTA). PTA, pure tone audiometry; HC, HearCheck.

## Table 2  Cross-tabulation of PTS versus HC test results

| | | PTS test results | | | |
|---|---|---|---|---|---|
| | | Refer | Pass | Missing | Total |
| *Impaired* | | | | | |
| HC test results | Refer | 136 | 2 | 1 | 139 |
| | Pass | 10 | 7 | 0 | 17 |
| | Missing | 0 | 0 | 0 | 0 |
| | Total | 146 | 9 | 1 | 156 |
| *Hearing* | | | | | |
| HC test results | Refer | 34 | 26 | 0 | 60 |
| | Pass | 45 | 340 | 0 | 385 |
| | Missing | 0 | 1 | 0 | 1 |
| | Total | 79 | 367 | 0 | 446 |
| *Missing* | | | | | |
| HC test results | Refer | 13 | 2 | 1 | 16 |
| | Pass | 3 | 2 | 1 | 6 |
| | Missing | 2 | 0 | 4 | 6 |
| | Total | 18 | 4 | 6 | 28 |

HC, HearCheck; PTS, pure tone screen.

in the calculation of sensitivity. The 445 hearing ears used in the calculation of specificity are highlighted in the second panel.

Table 3 reports the sensitivity and specificity of the screening tests. The sensitivity was 94.2% for PTS and 89.0% for HC. The 95% CI for sensitivity indicates that we can be fairly certain that the true sensitivity is no lower than 89% for PTS and 83% for HC. The McNemar's test result (p=0.02) indicates evidence that the true sensitivity is greater for PTS than for HC. The estimates of specificity were 82.2% for PTS and 86.5% for HC, with evidence provided by McNemar's test that the true specificity is higher for HC than PTS (p=0.02).

### False negatives

The mean hearing level across the four test frequencies on the PTA reference standard for the 19 ears that passed one or both of the screening tests but referred by the PTA was 28 (SD=9) dB compared with 48 (SD=21) dB for the remaining 136 impaired ears that referred on both PTS and HC. This indicates that impairment was less severe for the false negatives than the true positives.

## DISCUSSION

The main finding of our study is that PTS was better than HC with respect to sensitivity (5.2% in favour of PTS; 95% CI 0.2% to 10.1%; p=0.02), but inferior with respect to specificity (4.3% in favour of HC; 95% CI 0.4% to 8.2%; p=0.02).

The two-gate diagnostic test accuracy study design employed is widely acknowledged to be open to bias in the assessment of accuracy.[8] However, given the extremely low prevalence of hearing impairment in a school entry population, approximately 0.5%,[7] this was felt to be the only feasible design. In a traditional accuracy study where the test and reference standard are administered to all participants identified from a single source ('single gate') with no advance knowledge of their true disease status,[8] 16 000 school children in the UK would need to have been recruited to identify our target of 80 cases of hearing impairment and so offer the same precision for measuring sensitivity. The bias might lead to an overestimate of accuracy for each test individually, although we believe that it might have less impact on comparison of accuracy as both tests would be subject to any overestimation. Measuring PTS and HC accuracy in the near to ideal conditions in this study as opposed to the nosier circumstances that would prevail in schools is also likely to lead to inflation of accuracy of the tests individually.

We remain confident that there are no other studies that directly compare PTS with HC. Indeed there are very few directly comparative accuracy studies of any of the potential screening tests for hearing impairment.[7] Our findings are consistent with indirect comparison of PTS with other tests which suggest that PTS is superior.[3 7] What this study adds is that when PTS is used in a standard manner and HC is used in the manner designed by the manufacturers, there is a trade-off between sensitivity and specificity and a threshold effect may be part of the apparent difference between the two tests. However, given that thresholds are fixed, particularly for HC, it is reasonable to consider which of PTS and HC in the conventional forms used in the study would be preferable in practice. Some further insight into this is given by reflecting on the absolute numbers of false positives and false negatives when the differences in accuracy are applied to a population with a prevalence of hearing impairment similar to one which might be observed in practice. This is done in table 4 where the accuracy estimates are applied to a population of 10 000 with a prevalence of hearing impairment of 0.5% (ie, 50 with impairment). In most tests used for

## Table 3  Accuracy of PTS and HC

| Measure | Pure tone screen estimate (95% CI) | HearCheck estimate (95% CI) | Difference in accuracy (PTS – HC) | |
|---|---|---|---|---|
| | | | Estimate (95% CI) | p Value |
| Sensitivity | 94.2% (89.0% to 97.0%) | 89.0% (82.9% to 93.1%) | 5.2% (0.2% to 10.1%) | 0.02 |
| Specificity | 82.2% (77.7% to 86.0%) | 86.5% (82.5% to 90.0%) | −4.3% (−8.2% to −0.4%) | 0.02 |

HC, HearCheck; PTS, pure tone screen.

**Table 4** Frequency of test results per 10 000 screened in a hypothetical population

| Test results | Test | | Difference (PTS – HC) |
|---|---|---|---|
| | PTS | HC | |
| True positives | 47 | 45 | 2 |
| True negatives | 8179 | 8607 | |
| False positives | 1771 | 1343 | 428 |
| False negatives | 3 | 5 | |

HC, HearCheck; PTS, pure tone screen.

screening and triage, there is a preference for avoiding false negatives, because it may take many years for 'missed' individuals to re-engage with the health system, by which time the opportunity to successfully intervene may have been lost. However, as table 4 shows the number of false positives (1771 and 1343 for PTS and HC, respectively) is so much larger than the number of false negatives (3 and 5 for PTS and HC, respectively), that it is reasonable to question whether the cumulative added costs of unnecessary testing in false positives have reached a point where they outweigh the cumulative benefits of avoiding a much smaller number of false negatives. This is particularly true where the nature of the hearing impairment is milder in the missed cases than in those who correctly tested positive, as we found in this study. We did, however, note in another component study of this programme of work that the number of screened children attending for diagnostic evaluation was much less than would be implied by test specificity, suggesting strongly that the number of false positives in a screening programme is much less than would be indicated by test specificity in isolation.[7] This is because in a screening programme, those initially testing as impaired may have their screening result rechecked or reviewed before being finally sent for diagnostic evaluation. So the impact of false positives is overstated if one relies on test specificity in isolation rather than considering the specificity of the programme as a whole.

On balance, therefore, we retain the view that the reduced number of false negatives associated with PTS use (2 fewer per 10 000 children screened (table 4)) does outweigh the advantage in terms of test specificity apparently offered by HC which has 428 fewer false positives per 10 000 screened. The implications for practice are thus that where school entry hearing screening is still being used or is under consideration, PTS would be the better screening tool. We do note, however, that recently concerns have been expressed about the likely cost-effectiveness of SES relative to a system reliant on ad hoc identification of possible hearing impairment and referral for diagnostic evaluation, although this is an early finding needing confirmation.[7]

In terms of implications for research, while we note that this study gives robust information about the choice between PTS and HC, there are other alternative tests such as automated audiometry-based hearing screening

systems installed on laptops or hand-held devices,[9–14] otoacoustic emissions[15–19] and Automated Auditory Brainstem Response.[20] Although they have been the object of direct comparison of accuracy, further research is necessary to provide more robust evidence of their comparative performance, feasibility and cost-effectiveness in different country-specific contexts. Furthermore, we would suggest that if the arguments for the validity of comparative two-gate accuracy studies as used here are accepted this would be an appropriate and efficient means to evaluate relative accuracy in the future. Incorporating such direct comparisons into ongoing systematic reviews of single test accuracy studies should also be anticipated. Finally, the work we have done here on accuracy of the hearing screening tests should be extended to estimate the accuracy of the school entry hearing screening programme itself.

**Author affiliations**
[1]NIHR CLAHRC South West Peninsula (PenCLAHRC), University of Exeter Medical School, South Cloisters, St Luke's Campus, Exeter, UK
[2]Institute of Health Research, University of Exeter Medical School Luke's Campus, Exeter, UK
[3]NIHR Nottingham Biomedical Research Centre, University of Nottingham, Nottingham, UK
[4]Nottingham Audiology Services, Nottingham University Hospitals, Nottingham, UK
[5]Community Child Health, Ida Darwin Hospital, Fulbourn, Cambridge, UK
[6]Peninsula Clinical Trials Unit at Plymouth University (PenCTU), Plymouth University, Plymouth, UK
[7]Public Involvement Representative, Nottingham, UK

**Acknowledgements** We would like to thank the following: the participating families and schools; the audiologists at each centre for sending out invitations for the study; Venessa Vas for helping to complete the screening; Elliot Carter, Brendan Lane and Paul Williams from PenCTU for developing the data entry websites; Brian Wainman and Mark Warner from PenCTU for entering data and assisting with the data management tasks; former project team member Vasilis Nikolaou; members of and observers on the study steering committee (John Bamford, John Fitzgerald, Vicki Kirwin, Kevin Munro, Kate Northstone, Karen Smith, Eldon Spackman, Maria Koufali, Stacey Arland, Angela Shone); Jon Deeks for critical comments on the conduct and reporting of the study and Leala Watson for formatting images for the paper.

**Contributors** The protocol was developed and funding obtained by OCU, CH, RST, CB, JM and HF. HF was chief investigator with overall responsibility for the conduct of the study. OCU, CH, MO, ZZ, SE, RST, CB, JM, LC, JW and HF contributed to revisions of the design and conduct of the study. MO and SE managed the study and collected the diagnostic accuracy data. CB invited case children to the study. ZZ conducted an updated systematic review of the diagnostic accuracy of hearing screening tests. LC coordinated database design and development, data validation and data export. JW was the PPI representative on the study. OCU developed the statistical analysis plan which was critically revised by all authors. OCU undertook the analyses. OCU drafted the manuscript which was critically revised by all authors. OCU is the guarantor of the manuscript.

**Funding** The study was funded by the National Institute for Health Research (NIHR) Health Technology Assessment (HTA) programme (project no 10/63/03). OCU, ZZ, Jaime Peters and RST were supported by the NIHR Collaboration for Leadership in Applied Health Research and Care South West Peninsula (PenCLAHRC). CH was supported by the University of Exeter. MO and SE were supported by a grant from the NIHR HTA programme (10/63/03). CB was funded by Nottingham University Hospitals National Health Service (NHS) Trust. JM was funded by CCS NHS Trust and supported by Addenbrookes Hospital, Cambridge University Hospitals Foundation Trust. HF was funded by the University of Nottingham and supported by the NIHR Nottingham Hearing Biomedical Research Unit.

**Disclaimer** The views and opinions expressed therein are those of the authors and do not necessarily reflect those of the HTA programme, the NHS, the NIHR or the Department of Health. Role of the study sponsor and funder: Neither the funding

body (NIHR HTA programme) nor the sponsor (University of Nottingham) had a role in the design of the study, collection, analysis and interpretation of the data; writing of the paper or the decision to submit it for publication. Independence of researchers from funder: All researchers worked independently from the funder.

**Competing interests** None declared.

**Patient consent** Parent/guardian consent obtained.

**Ethics approval** Ethical approval was granted by the West Midlands, Staffordshire Research Ethics Committee (Ref: 106333).

**Provenance and peer review** Not commissioned; externally peer reviewed.

**Data sharing statement** Participant level data, the full dataset and statistical code are available from the corresponding author. Consent for this was not obtained but the presented data are anonymised and risk of identification is low.

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
