## [Reviewer comments · BMJ Open]

ARTICLE DETAILS

TITLE (PROVISIONAL)	A directly comparative two-gate case-control diagnostic accuracy study of the pure tone screen and HearCheck Screener tests for identifying hearing impairment in school children
AUTHORS	Ukoumunne, Obioha; Hyde, Chris; Ozolins, Mara; Zhelev, Zhivko; Errington, Sam; Taylor, Rod; Benton, Claire; Moody, Joanne; Cocking, Laura; Watson, Julian; Fortnum, Heather

VERSION 1 - REVIEW

REVIEWER	ROBERTO MIQUELINO DE OLIVEIRA BECK UNIVERSITY OF SAO PAULO - SCHOOL OF MEDICINE SAO PAULO - BRAZIL
REVIEW RETURNED	01-May-2017

GENERAL COMMENTS	This draft compared two methods for hearing screening in children between 4-6 year-old. We know that late-onset hearing loss is possible and children at this age can have great problems with any unidentified impairment. I have just some questions that could improve your paper: - I couldn't understand why hearing loss was considered when threshold was > 30 dB and normal hearing when < 20 dB?- How conductive hearing loss was classified or identified? This is an important aspect in this age group- About HC screener use, how does it function? There is a large explanation in the appendix section, but maybe it will not be published.- Table 2 needs more explanation (captions)
--

REVIEWER	Stavros Hatzopoulos Audiology, University of Ferrara, Italy
REVIEW RETURNED	09-May-2017

GENERAL COMMENTS	Excellent organization and presentation of the material. The Research question is answered in detailed and the readers of the Journal can find a new point of reference in this manuscript.
---

VERSION 1 – AUTHOR RESPONSE

Responses to Reviewer 1 comments:

We thank Reviewer 1 for these comments.

1) For the reference standard, hearing impairment was defined as present when the threshold was >=

30dB and absent when <30dB. This is the usual manner in which impairment is defined based on the PTA reference standard. For example, see Orlando MS and Frank T. Audiometer and AudioScope hearing screening compared with threshold test in young children. Journal of Pediatrics. 1987; 110: 261-263

2) The screening tests and reference standard as administered in this study were never intended to discriminate between sensorineural hearing loss and conductive hearing loss. This sub-classification was only established after those who failed the study test were invited for testing in Audiology, and we do not have access to these data. Most of those with a hearing impairment were recruited as intended cases based on the fact that they had an established impairment. We would, therefore, expect the majority of these to have sensorineural hearing loss rather than conductive hearing loss.

3) Regarding the function of the HearCheck test we have added a website address to the main text (at the end of the second paragraph in the Introduction section) that the readers can refer to: <http://www.connevans.co.uk/product/2831233/38SHEARCHECK/Siemens-HearCheck-Screener> [accessed 18th May 2017]

4) It is my understanding that the material in the Appendix will be made available by BMJ Open but we will take advice from the editor on this.

5) We agree with the reviewer that further information is required to describe the contents of Table 2. We have added this to the main text (see Sensitivity and Specificity sub-section in the Results section).

Responses to Reviewer 2 comments:

We thank Reviewer 2 for these comments.

VERSION 2 – REVIEW

REVIEWER	Roberto Miquelino de Oliveira Beck University of Sao Paulo school of Medicine
REVIEW RETURNED	08-Jun-2017

GENERAL COMMENTS	The authors answered and corrected all the questions of the first review.
---